

# Changes in the spectrum of kidney diseases: a 14-year single-center retrospective renal biopsy study from southeast China

Zishan Lin[1,2,3,*], Shidong Xie[1,2,3,*], Wenfeng Wang[1,2,3], Tao Hong[1,2,3], Bingjing Jiang[1,2,3], Caiming Chen[1,2,3], Xuan Tao[4], Dewen Jiang[1,2,3], Jianxin Wan[1,2,3], Hong Chen[4] and Yanfang Xu[1,2,3]

[1] Department of Nephrology, Blood Purification Research Center, the First Affiliated Hospital, Fujian Medical University, Fuzhou, China
[2] Research Center for Metabolic Chronic Kidney Disease, the First Affiliated Hospital, Fujian Medical University, Fuzhou, China
[3] Department of Nephrology, National Regional Medical Center, Binhai Campus of the First Affiliated Hospital, Fujian Medical University, Fuzhou, China
[4] Department of Pathology, the First Affiliated Hospital, Fujian Medical University, Fuzhou, China
[*] These authors contributed equally to this work.

Corresponding author
Yanfang Xu, xuyanfang99@hotmail.com

## ABSTRACT

**Objective**. Kidney disease has become a significant public health issue. Socioeconomic improvements and environmental changes in China have influenced the kidney disease spectrum. This study aims to examine the changing spectrum of biopsy-confirmed kidney diseases in a single center from southeastern China and explore their clinical-pathological correlations.

**Methods**. Patients who underwent renal biopsy at the First Affiliated Hospital of Fujian Medical University from January 2010 to December 2023 were included in this retrospective study. Clinicopathological data were collected and analyzed.

**Results**. 2,832 patients were enrolled. Primary glomerular diseases accounted for the majority of kidney diseases (69.0%), followed by secondary glomerular diseases, tubulointerstitial diseases, and others. Among primary glomerular diseases, membranous nephropathy (MN) was the most prevalent (34.9%), followed closely by IgA nephropathy (IgAN) (34.5%) and minimal change disease (18.5%), with an increasing trend for MN. For secondary glomerular diseases, lupus nephritis (34.3%), diabetic nephropathy (DN) (24.8%), and hepatitis B virus-associated nephropathy (HBVN) (9.0%) were the most common. HBVN showed a significant decrease, while DN increased annually. Nephrotic syndrome (NS) was the most frequent indication for renal biopsy (40.3%), followed by progressive chronic kidney disease (22.4%). MN (47.2%) was the leading pathology in NS, and IgAN (51.6%) was predominant in patients with proteinuria and hematuria.

**Conclusions**. The spectrum of kidney disease has changed within the last 14 years. The relative frequency of MN and DN increased significantly, while that of HBVN decreased significantly. These findings highlight the need for ongoing public health efforts tailored to the changing spectrum of kidney diseases.

## INTRODUCTION

Chronic kidney disease (CKD) has become a critical global public health issue, with rising prevalence, poor outcomes, and increasing costs demanding urgent attention. In China, the prevalence of CKD has grown from 6.7% to 10.6% over the past three decades, and mortality rates have increased from 8.3 to 13.8 per 100,000 people per year (*Li et al., 2022*). Early detection and timely treatment of kidney diseases are crucial in preventing the progression to advanced CKD (*Dirks et al., 2004*; *Bello, Nwankwo & El Nahas, 2005*). Among kidney diseases, membranous nephropathy (MN) and IgA nephropathy (IgAN) are the most prevalent (*Xu et al., 2016a*). Treatments and prognoses differ significantly among the various types of kidney diseases. While certain laboratory predictors, such as anti-PLA2R antibodies, serve as reliable markers for idiopathic MN, renal biopsy remains the gold standard for diagnosing a broad spectrum of kidney diseases (*Zhang & Parikh, 2019*; *van de Logt et al., 2019*).

Kidney disease prevalence varies by region. For example, IgAN predominates in Asia and Europe, while focal segmental glomerulosclerosis (FSGS) is more common in the US and Brazil (*Zhou et al., 2009*; *Zhang et al., 2014*; *Rychlík et al., 2004*; *Braden et al., 2000*; *Polito, de Moura & Kirsztajn, 2010*). Studies also indicate a global shift in the spectrum of kidney diseases. For instance, in Denmark, the last three decades have witnessed a consistent rise in the incidence of FSGS and anti-neutrophil cytoplasmic antibody (ANCA)-associated vasculitis (*Heaf, Sørensen & Hansen, 2020*). In China, the disease spectrum has changed due to factors like hepatitis B, diabetes, hypertension, population aging, and environmental shifts (*Zhou et al., 2009*; *Zhang et al., 2014*; *Yang et al., 2018*; *Pan et al., 2013*). These changes have led to increased cases of diabetic nephropathy (DN) and hypertensive nephropathy (HTN) (*Yang et al., 2020*; *Bu et al., 2024*). Variability in disease trends has been reported by different centers, reflecting considerable regional differences. For instance, a central China study from 1994 to 2014 reported rising incidences of IgAN, MN, and minimal change disease (MCD), while southern China saw increases in MN and DN from 2010 to 2018 (*Xu et al., 2016a*; *Zheng et al., 2022*). Furthermore, regarding the clinical indications for renal biopsy, nephrotic syndrome (NS) is the most common indication for renal biopsy in northern China, while proteinuria and hematuria predominante in central China (*Xu et al., 2016a*; *Zhou et al., 2009*; *Wu et al., 2011*).

However, limited data exist on the recent changes in kidney disease spectra confirmed by renal biopsy in southeastern China. This study retrospectively analyzed renal biopsy data from the First Affiliated Hospital of Fujian Medical University from 2010 to 2023 to investigate these changes and explore their clinical-pathological correlations.

## METHODS

### Study population

This study included patients who underwent renal biopsy at the First Affiliated Hospital of Fujian Medical University from January 2010 to December 2023. Our hospital is a tertiary referral center located in the provincial capital of southeastern China, serving a population of approximately 8.0 million residents from both the province and surrounding areas. This study was performed in accordance with the Helsinki Declaration and approved by the ethics committee of the First Affiliated Hospital of Fujian Medical University [2015]084-2. Written informed consent was obtained from each patient.

Exclusion criteria included fewer than 10 glomeruli available for light microscopy (23 patients), prior kidney transplantation (two patients), repeat biopsies (three patients), age under 14 years (10 patients), or incomplete demographic or clinical data (11 patients).

Indications for kidney biopsy were as follows: (1) NS; (2) hematuria; (3) proteinuria; (4) both proteinuria and hematuria; (5) acute kidney injury (AKI); (6) NS combined with AKI (NS+AKI); (7) progressive CKD; and (8) others, such as renal involvement in systemic diseases. Patients with CKD combined with NS were categorized under NS, and those with CKD combined with acute kidney injury (AKI) were categorized under AKI. The indications for renal biopsy remained consistent throughout the study period. All patients met established indications for the procedure.

### Data collection and definition

Demographic information, laboratory data, clinical diagnoses, and renal pathological findings were collected.

Clinical manifestations were defined as follows: (1) NS was defined as more than 3.5 g of proteinuria in a 24-hour urine sample and serum albumin below 30 g/L; (2) AKI was characterized by a serum creatinine increase of at least 26.5 $\mu$mol/L within 48 h, a rise to 1.5 times baseline within 7 days, or urine output below 0.5 mL/kg/hour for more than 6 h (*Kidney Disease: Improving Global Outcomes (KDIGO) Acute Kidney Injury Work Group, 2012*); (3) Progressive CKD was defined by persistent kidney damage or worsening kidney function over a period exceeding 3 months, confirmed by tests of kidney function, urine analysis, and imaging.

### Pathological examination and diagnosis

All renal biopsies were performed under ultrasound guidance. Specimens were prepared for light microscopy, electron microscopy, and immunofluorescence. Tissue samples were fixed in 10% formalin, embedded in paraffin, and sectioned into two $\mu$m slices. Staining included hematoxylin and eosin, periodic acid-Schiff, periodic acid-silver-methenamine, Masson's trichrome, and Congo red. For immunofluorescence, frozen tissues were cut into five $\mu$m sections and directly stained with fluorescein isothiocyanate (FITC)-labeled antibodies (IgG, IgA, IgM, C3, C4, C1q, and FRA). Indirect immunofluorescence was performed as an auxiliary diagnostic tool for patients suspected of having hepatitis B virus-associated nephropathy (HBVN). This involved using FITC-conjugated mouse monoclonal antibody F(ab')2 fragments to detect the deposition of HBsAg, HBcAg, and HBeAg in frozen sections.

The deposition patterns and intensity were assessed under a fluorescence microscope in a dark environment. All patients also underwent electron microscopy. The pathological diagnoses were independently reviewed by two experienced nephropathologists based on histopathological features and the patients' clinical data, including clinical manifestations and laboratory test results. In cases of diagnostic discrepancies, a third nephropathologist was consulted for confirmation.

The pathological classification of renal diseases in this study followed the revised 1995 World Health Organization criteria, as detailed in Table S1 (*Churg & Glassock, 1995*; *Weening et al., 2004*). Kidney diseases were grouped into four main categories: (1) primary glomerulonephritis (PGN), including but not limited to MN, MCD, IgAN, FSGS, mesangial proliferative glomerulonephritis, membranoproliferative glomerulonephritis, and endocapillary proliferative glomerulonephritis; (2) secondary glomerulonephritis (SGN), including but not limited to lupus nephritis (LN), HBVN, DN, Henoch-Schönlein purpura nephritis (HSPN), and anti-neutrophil cytoplasmic antibody (ANCA)-associated glomerulonephritis; (3) tubulointerstitial nephritis (TIN), including but not limited to acute tubular necrosis and acute interstitial nephritis; and (4) other renal diseases, including but not limited to genetic kidney diseases like Alport syndrome, thin basement membrane nephropathy, Fabry disease, and rare kidney diseases such as lipoprotein nephropathy. In this study, patients were classified as having PGN only after excluding secondary factors, such as HBVN, based on clinical data and pathological findings. Those with glomerulonephritis that could be unequivocally attributed to secondary causes were classified as having SGN.

## Statistical analysis

SPSS version 25.0 and GraphPad Prism 9.5 were employed for data analysis and graphing. Percentages were used to describe categorical variables. Proportions of major pathological types were calculated for different periods. The distribution of major pathological types across different time periods was analyzed using $\chi 2$ test for trend. Fisher's exact test was applied when the expected frequency was less than five. All tests were two-tailed, and *P* values less than 0.05 were considered statistically significant.

## RESULTS

### The pathological and clinical spectrum of kidney disease in total

This retrospective study involved 2,832 patients who underwent renal biopsies at the First Affiliated Hospital of Fujian Medical University between January 2010 and December 2023. The annual number of renal biopsies showed a consistent upward trend, increasing from 81 in 2010 to 381 in 2023 (Fig. 1A). Data were divided into four periods: 2010–2012 (Period 1), 2013–2016 (Period 2), 2017–2020 (Period 3), and 2021–2023 (Period 4), with patients stratified into age groups of 14–24, 25–44, 45–59, and $\geq$ 60 years at the time of biopsy. As shown in Table 1, the number of patients increased from 290 in Period 1 to 1,094 in Period 4. The cohort had a male-to-female ratio of 1.28:1, with males comprising 56.1% of the participants. Notably, the proportion of patients aged $\geq$ 60 years increased from 9.3%

**Table 1  Characteristics of patients over different periods from 2010 to 2023.**

| | 2010–2012 (n = 290) | 2013–2016 (n = 536) | 2017–2020 (n = 912) | 2021–2023 (n = 1,094) | $\chi^2$ | P-value | Total (n = 2,832) |
|---|---|---|---|---|---|---|---|
| **Sex** | | | | | | | |
| Male, n (%) | 163 (56.2%) | 284 (53.0%) | 533 (58.4%) | 608 (55.6%) | 0.155 | 0.693 | 1588 (56.1%) |
| Female, n (%) | 127 (43.8%) | 252 (47.0%) | 379 (41.6%) | 486 (44.4%) | 0.155 | 0.693 | 1244 (43.9%) |
| **Age category** | | | | | | | |
| 14–24 years, n (%) | 69 (23.8%) | 86 (16.1%) | 93 (10.2%) | 125 (11.5%) | 28.937 | <0.001 | 373 (13.2%) |
| 25–44 years, n (%) | 131 (45.2%) | 217 (40.5%) | 333 (36.5%) | 393 (35.9%) | 9.179 | 0.002 | 1,074 (37.9%) |
| 45–59 years, n (%) | 63 (21.7%) | 139 (25.9%) | 269 (29.5%) | 322 (29.4%) | 7.090 | 0.008 | 793 (28.0%) |
| ≥60 years, n (%) | 27 (9.3%) | 94 (17.5%) | 217 (23.8%) | 254 (23.2%) | 26.509 | <0.001 | 592 (20.9%) |
| **Pathologic type** | | | | | | | |
| PGN, n (%) | 205 (70.6%) | 375 (70.0%) | 638 (70.0%) | 738 (67.5%) | 1.697 | 0.193 | 1956 (69.0%) |
| MN, n (%) | 40 (19.5%) | 125 (33.3%) | 267 (41.8%) | 250 (33.9%) | 8.585 | 0.003 | 682 (34.9%) |
| MCD, n (%) 34 | (16.6%) | 74 (19.7%) | 101 (15.8%) | 152 (20.6%) | 1.199 | 0.274 | 361 (18.5%) |
| IgAN, n (%) | 84 (41.0%) | 122 (32.5%) | 213 (33.4%) | 255 (34.6%) | 0.810 | 0.368 | 674 (34.5%) |
| Other PGN, n (%) | 47 (22.9%) | 54 (14.5%) | 57 (9.0%) | 81 (10.9%) | 18.100 | <0.001 | 239 (12.1%) |
| SGN, n (%) | 77 (26.6%) | 144 (26.9%) | 245 (26.9%) | 323 (29.5%) | 1.728 | 0.189 | 789 (27.9%) |
| LN, n (%) | 30 (39.0%) | 63 (43.8%) | 64 (26.1%) | 114 (35.3%) | 1.769 | 0.183 | 271 (34.3%) |
| HBVN, n (%) | 19 (24.7%) | 29 (20.1%) | 20 (8.2%) | 3 (0.9%) | 69.140 | <0.001 | 71 (9.0%) |
| DN, n (%) | 6 (7.8%) | 16 (11.1%) | 78 (31.8%) | 96 (29.7%) | 26.379 | <0.001 | 196 (24.8%) |
| Other SGN, n (%) | 22 (28.5%) | 36 (25.0%) | 83 (33.9%) | 110 (34.1%) | 2.891 | 0.089 | 251 (31.9%) |
| TIN, n (%) | 4 (1.4%) | 8 (1.5%) | 16 (1.7%) | 15 (1.4%) | 0.011 | 0.918 | 43 (1.5%) |
| Others, n (%) | 4 (1.4%) | 9 (1.6%) | 13 (1.4%) | 18 (1.6%) | 0.042 | 0.837 | 44 (1.6%) |
| **Clinical indication** | | | | | | | |
| NS, n (%) | 110 (37.9%) | 226 (42.2%) | 392 (43.0%) | 412 (37.7%) | 0.942 | 0.332 | 1140 (40.3%) |
| NS+AKI, n (%) | 8 (2.8%) | 31 (5.8%) | 41 (4.5%) | 48 (4.4%) | 0.033 | 0.856 | 128 (4.5%) |
| AKI, n (%) | 7 (2.4%) | 23 (4.3%) | 34 (3.7%) | 36 (3.3%) | 0.001 | 0.982 | 100 (3.5%) |
| Progressive CKD, n (%) | 55 (19.0%) | 81 (15.1%) | 188 (20.6%) | 310 (28.3%) | 31.671 | <0.001 | 634 (22.4%) |
| Prot, n (%) | 19 (6.6%) | 31 (5.8%) | 58 (6.4%) | 74 (6.8%) | 0.245 | 0.621 | 182 (6.4%) |
| Hem, n (%) | 10 (3.4%) | 13 (2.4%) | 8 (0.9%) | 11 (1.0%) | 11.520 | <0.001 | 42 (1.5%) |
| Prot+Hem, n (%) | 77 (26.6%) | 125 (23.3%) | 180 (19.7%) | 192 (17.5%) | 15.106 | <0.001 | 574 (20.3%) |
| Others, n (%) | 4 (1.4%) | 6 (1.1%) | 11 (1.2%) | 11 (1.0%) | 0.243 | 0.622 | 32 (1.1%) |

**Notes.**

Data were expressed as number (%).

PGN, primary glomerulonephritis; MN, membranous nephropathy; CKD, chronic kidney disease; IgAN, IgA nephropathy; SGN, secondary glomerulonephritis; LN, lupus nephritis; HBVN, hepatitis B virus-associated nephropathy; DN, diabetic nephropathy; NS, nephrotic syndrome; AKI, acute kidney injury; Prot, proteinuria; Hem, hematuria.

in Period 1 to 23.8% in Period 3 ($\chi^2$ = 26.509, P < 0.001), indicating a demographic shift toward an older population.

PGN, SGN, TIN, and other pathologies accounted for 69.0%, 27.9%, 1.5%, and 1.6% of diagnoses, respectively. PGN was the most common diagnosis across all periods, representing 70.6%, 70.0%, 70.0%, and 67.5% of cases in Periods 1 to 4, respectively (Fig. 1B). SGN increased gradually, with proportions of 26.6%, 26.9%, 26.9%, and 29.5%, respectively. Figure 1C shows that NS was the leading indication for renal biopsy (40.3%), followed by progressive CKD (22.4%) and proteinuria + hematuria (20.3%). Isolated

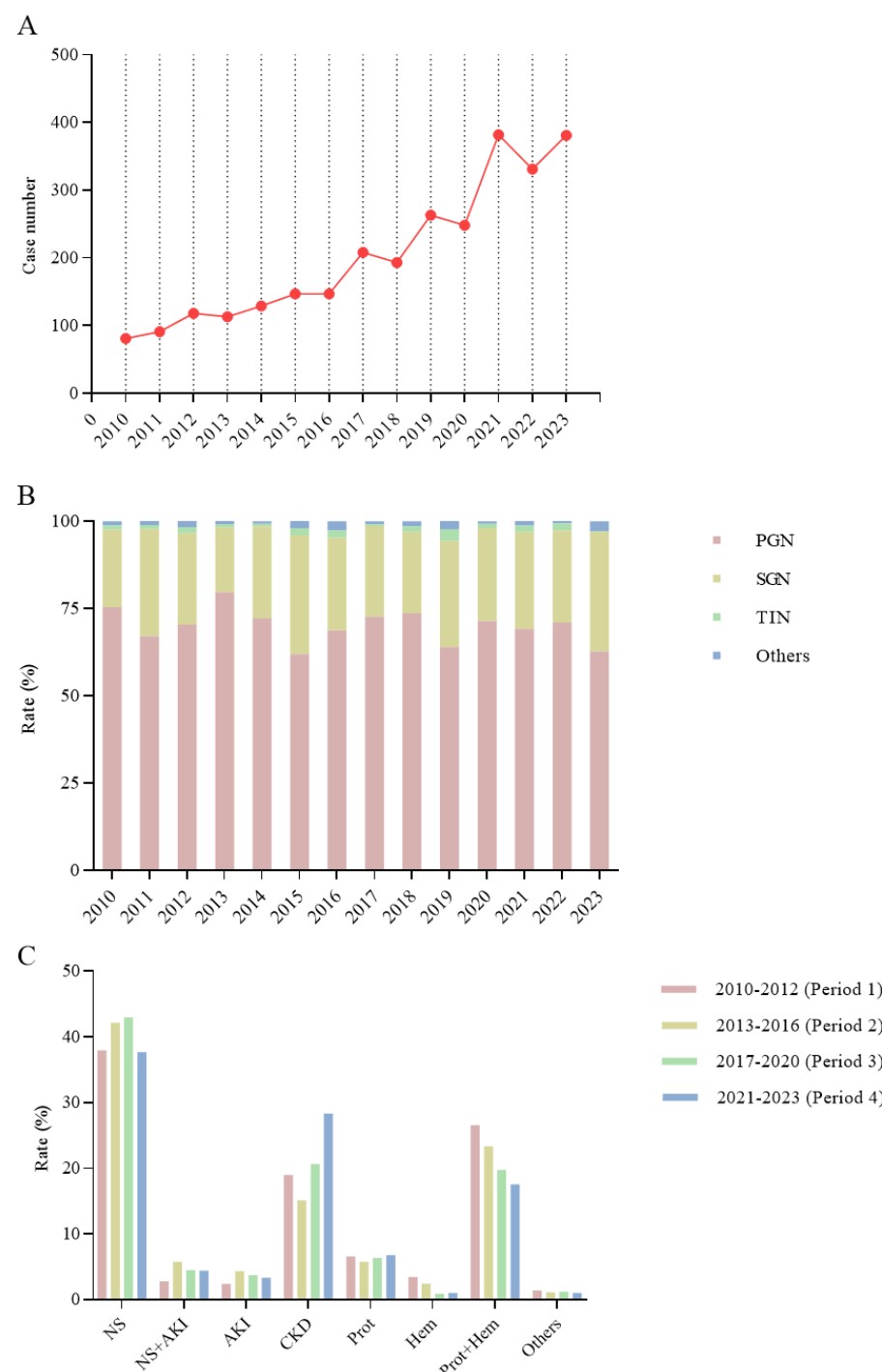

**Figure 1** (A) Trends in the number of renal biopsy cases from 2010 to 2023. (B) Changes in the spectrum of kidney diseases from 2010 to 2023. (C) Changes in renal biopsy indications across four periods.

proteinuria accounted for 6.4% of cases, NS+AKI for 4.5%, AKI alone for 3.5%, and isolated hematuria for 1.5%. The proportion of patients with both proteinuria and hematuria decreased significantly over time ($\chi^2 = 15.106$, $P < 0.001$), from 26.6% in Period 1 to 17.5% in Period 4. Conversely, progressive CKD cases increased from 19.0% in Period 1 to 28.3% in Period 4 ($\chi$ (*Dirks et al., 2004*) $= 31.671$, $P < 0.001$).

**The pathological spectrum and demographic features of PGN**
The three most common subtypes of PGN were MN, MCD, and IgAN, comprising 34.9%, 18.5%, and 34.5% of cases, respectively (Fig. 2A). From Period 1 to Period 4, the proportion of MN increased significantly, rising from 19.5% in Period 1 to 33.9% in Period 4 ($\chi^2 = 8.585$, $P = 0.003$), with a peak of 41.8% observed in Period 3. In contrast, the proportion of IgAN was highest in Period 1 (41.0%), but decreased sharply to 32.5% in Period 2. However, this change was not statistically significant ($\chi^2 = 0.810$, $P = 0.368$). During Periods 2 and 3, MN was the most common PGN subtype, whereas IgAN was the dominant subtype in Periods 1 and 4. Over the study period, the proportion of MCD remained relatively stable, without showing any statistically significant changes ($\chi^2 = 1.199$, $P = 0.274$).

As shown in Fig. 3A, MN, MCD, and IgAN displayed significant age-related variation. MN proportion increased with age, accounting for 11.3% of cases in the 14–24 years age group and 63.8% in the $\geq 60$ years group ($\chi^2 = 755.020$, $P < 0.001$). IgAN was more frequent in younger individuals, peaking at 54.5% of PGN cases in the 25–44 years group, but decreasing to 8.7% in the $\geq 60$ years group. MCD peaked at 36.5% in the 14–24 years group and declined with age.

Analysis of sex ratios across different age groups (Figs. 3B–3D) revealed that men were more frequently diagnosed with MN and IgAN compared to women across all age groups. MCD also showed a higher proportion among men, particularly in the 14–24 years group.

**The pathological spectrum and demographic features of SGN**
As shown in Fig. 2B, LN, DN, and HBVN were the leading causes of SGN, accounting for 34.3%, 24.8%, and 9.0% of all SGN cases, respectively.

From Period 1 to Period 4, the proportion of DN increased significantly, rising from 7.8% in Period 1 to 29.7% in Period 4 ($\chi^2 = 26.379$, $P < 0.001$). In Period 3, DN became the predominant SGN subtype, though its proportion decreased slightly afterward. Meanwhile, HBVN exhibited a steep decline, dropping from 24.7% in Period 1 to 0.9% in Period 4 ($\chi^2 = 69.140$, $P < 0.001$). From Period 1 to Period 2, LN remained the most prevalent SGN subtype and re-emerged the dominant subtype in Period 4 (35.3%), despite experiencing a notable decrease from 43.8% in Period 2 to 26.1% in Period 3.

Age-related trends for LN and DN are shown in Fig. 4A. LN was more commonly diagnosed in younger patients, accounting for 71.7% of cases in the 14–24 years group, but this dropped to 8.3% in those aged $\geq 60$ years. Conversely, DN was more frequently observed in older patients, with the highest proportion of 43.8% in the 45–59 years group, followed by a slight decline to 34.8% in those aged 60 years and older. There were no significant age-related variations for HBVN during the study period.

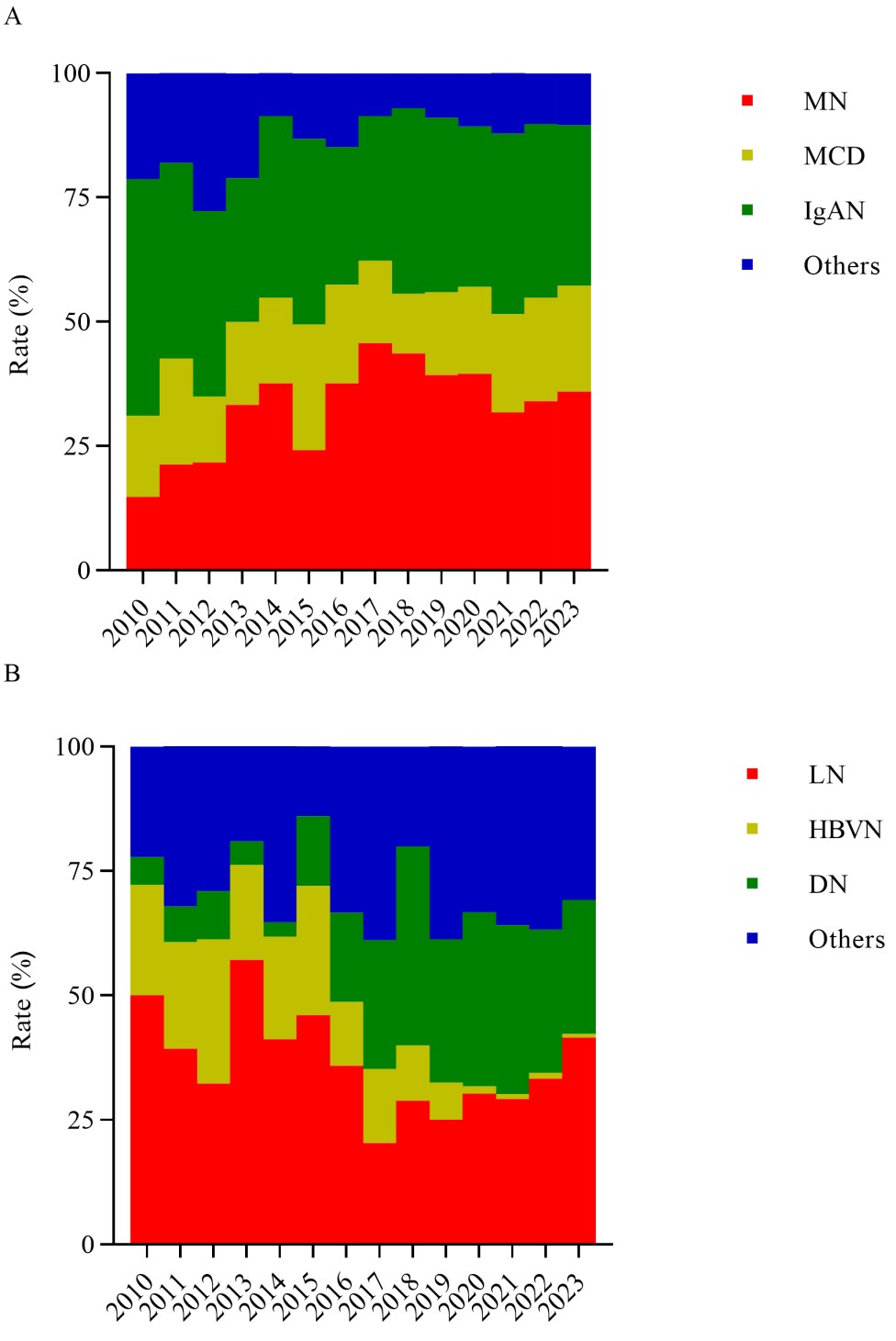

**Figure 2** Frequency of specific primary glomerular diseases (A) and secondary glomerular diseases (B) over the study period.

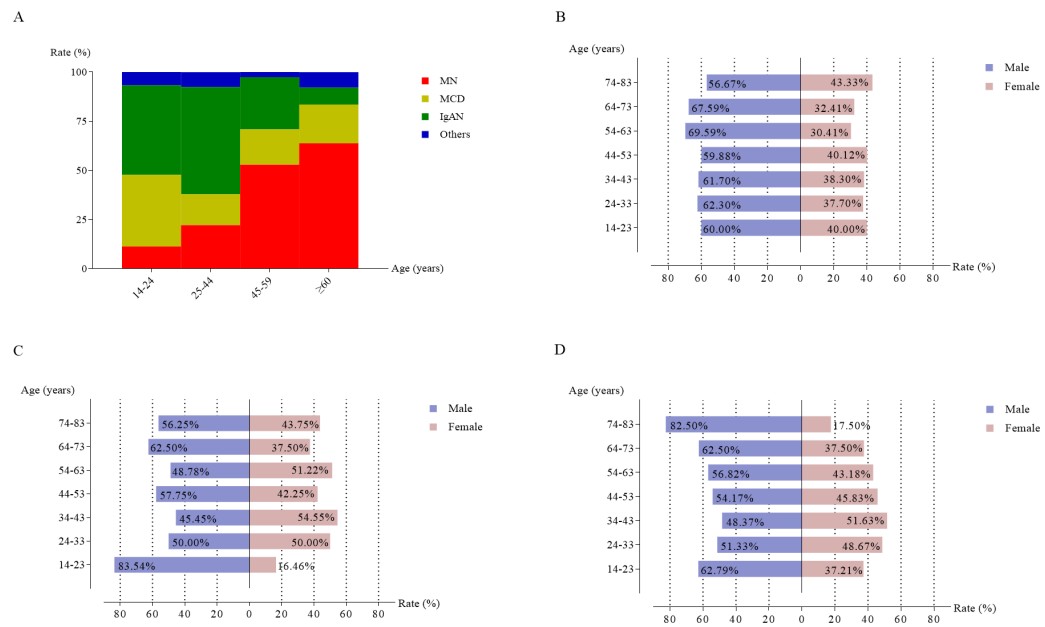

**Figure 3** The frequency of the most common histological categories in primary glomerular nephropathy based on different age groups (A). Gender distribution of membranous nephropathy (B), minimal change disease (C), and IgA nephropathy (D) across different age group.

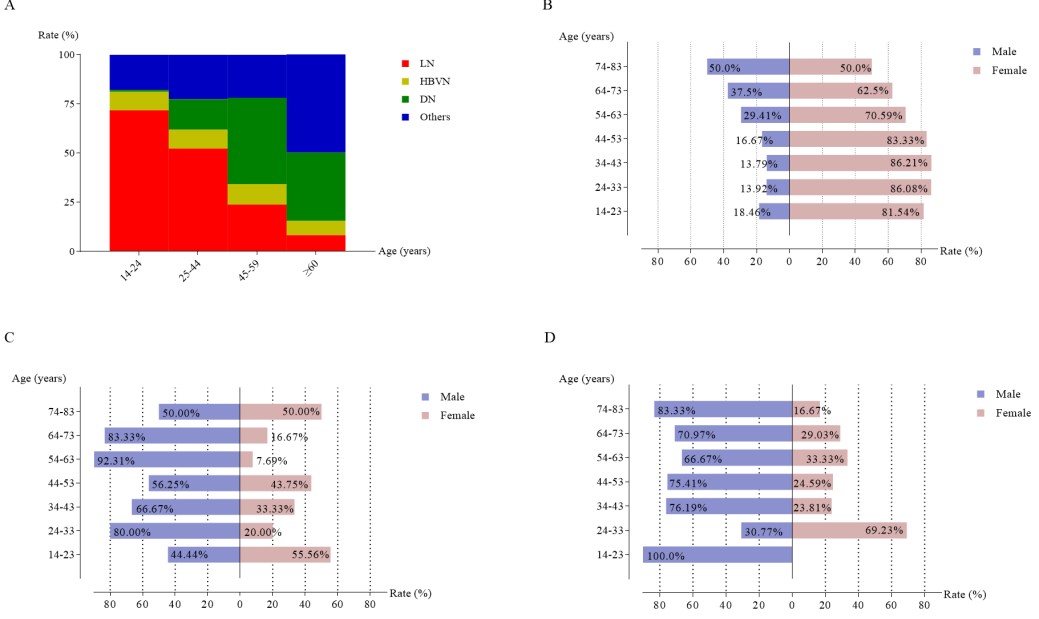

**Figure 4** The frequency of the most common histological categories in secondary glomerular nephropathy based on different age groups (A). Gender distribution of lupus nephritis (B), hepatitis B virus-associated glomerulonephritis (C), and diabetic nephropathy (D).

Figures 4B–4D show that LN was more common in women across all age groups, while DN was more prevalent in men aged 34–83 years. No significant gender-based differences were observed for HBVN across age categories.

## Changes in clinicopathologic correlations

Among patients undergoing renal biopsy at our center, NS was the most frequent indication (40.3%), followed by progressive CKD (22.4%) and proteinuria + hematuria (20.3%). As shown in Fig. 5A, MN was the most common pathology among NS patients, accounting for 47.2% of cases, followed by MCD (17.4%), LN (8.0%), and DN (7.3%). The proportion of MN in NS patients increased significantly, from 27.8% in 2010–2011 to 49.3% in 2022–2023 ($\chi^2 = 8.990$, $P = 0.003$). MN remained the predominant NS subtype across all periods (Fig. 5B). The proportion of DN among NS patients also increased significantly ($\chi^2 = 15.536$, $P < 0.001$), rising from 1.4% in 2010–2011 to 9.4% in 2022–2023.

For patients with proteinuria + hematuria, IgAN was the most prevalent diagnosis, comprising 51.6% of cases, followed by LN (12.4%), MN (11.8%), and HSPN (4.0%) (Fig. 5C). IgAN consistently remained the most common renal disease among patients with proteinuria + hematuria, peaking at 60.3% in 2020–2021 (Fig. 5D).

Among patients with progressive CKD, the most prevalent renal pathology was IgAN (35.33%), followed by DN (12.78%), LN (7.89%), and MCD (5.99%) (Fig. 5E). IgAN remained the most common pathology across all periods in patients presenting with progressive CKD (Fig. 5F).

## DISCUSSION

Over the past few decades, the spectrum of kidney diseases has undergone substantial changes. To better understand the evolution, we analyzed the clinical and pathological data of 2,832 patients who underwent renal biopsy from the First Affiliated Hospital of Fujian Medical University between 2010 and 2023. During this period, the number of renal biopsies increased significantly, driven by improved public health awareness, socioeconomic advancements, and enhancements in both the safety and technical proficiency of renal biopsy procedures (*Torres Muñoz et al., 2011*). Additionally, the proportion of patients aged 60 and above increased, not only reflecting the aging population in China but also due to improved healthcare access, advances in diagnostic techniques, and a higher prevalence of age-related conditions such as diabetes and hypertension (*Woo et al., 2002*; *Yang et al., 2010*). These findings align with reports from other regions of the country (*Zhou et al., 2009*; *Pan et al., 2013*).

In our study, PGN accounted for 69.0% of all renal biopsy cases, making it the most prevalent type of kidney disease. However, its proportion gradually decreased over time, while SGN showed a steady increase. This trend is consistent with findings from other regions in China and internationally. For example, SGN, such as DN and hypertensive nephropathy, has risen significantly in the US and Brazil due to the increasing prevalence of diabetes and hypertension (*Zhou et al., 2009*; *Zhang et al., 2014*; *Polito, de Moura & Kirsztajn, 2010*; *Pan et al., 2013*; *Burrows, Koyama & Pavkov, 2022*).
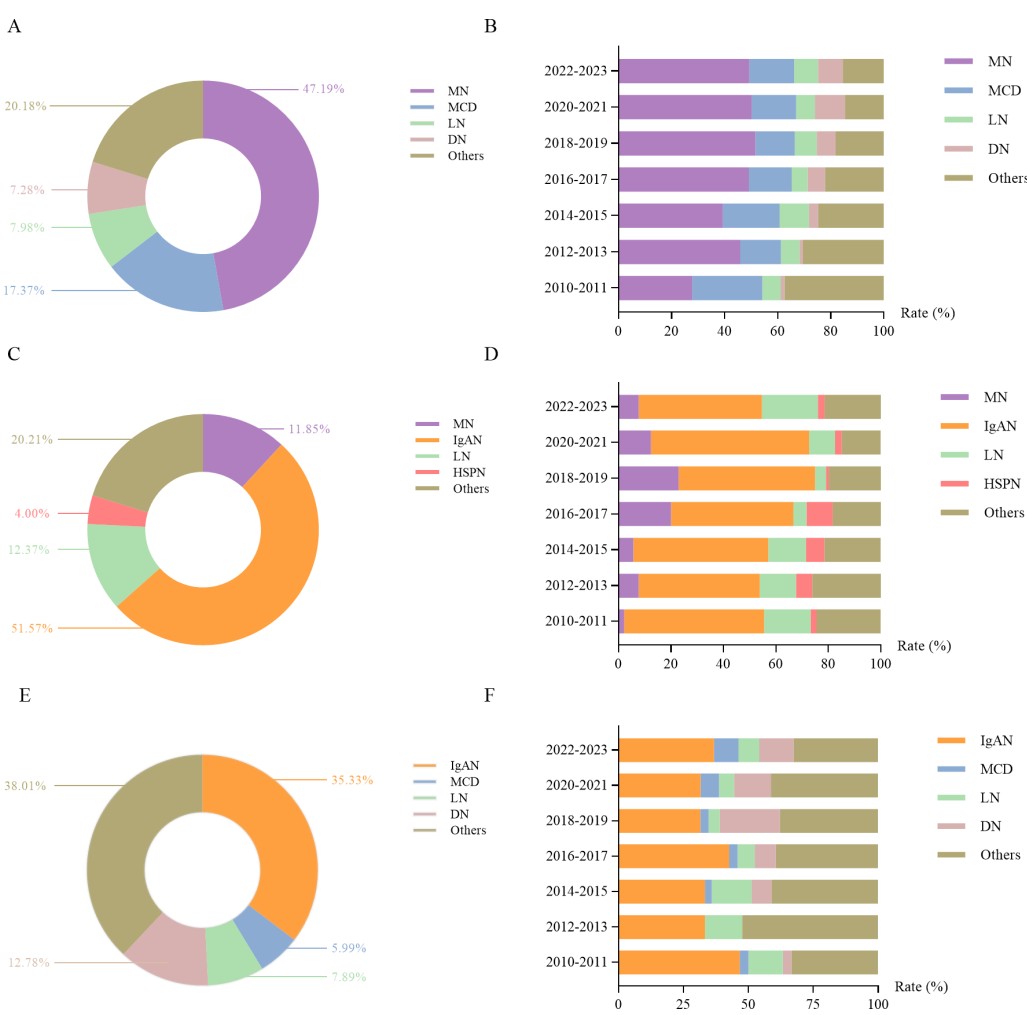

**Figure 5  Distribution of different kidney diseases among patients with nephrotic syndrome and their changes over time (A, B).** Distribution of different kidney diseases among patients with combined proteinuria and hematuria and their changes over time (C, D). Distribution of different kidney diseases among patients with progressive CKD and their changes over time (E, F).

This shift is likely driven by the rising prevalence of systemic diseases, which affect multiple organs including kidney.

MN emerged as the most prevalent type of PGN at our center, accounting for 34.9% of cases. It was also the most common pathological type among NS patients, with its proportion rising from 14.8% in 2010 to 36.0% in 2023. This upward trend is consistent with studies from other regions of China and internationally, including Spain and India (*Zhang et al., 2014*; *Pan et al., 2013*; *Xu et al., 2016b*; *Rivera et al., 2002*; *Das, Dakshinamurty & Prayaga, 2011*). However, contrasting trends have been observed in countries such as Japan, South Korea, the US and the UK, where the prevalence of MN has been declining (*Braden et al., 2000*; *Sugiyama et al., 2011*; *Chang et al., 2009*; *Hanko et al., 2009*). These differences may be attributed to variations in genetic predisposition, environmental

exposure, and kidney disease screening policies. In recent years, urine routine tests have become a standard component of health screenings in schools and workplaces. This has led to increased attention to the importance of anti-PLA2R antibody screening for patients with abnormal urine test results, which has likely contributed to the improved detection rate of MN. Additionally, exposure to high levels of PM2.5 (fine particulate matter) has been strongly linked to an increased risk of MN (*Xu et al., 2016b*; *Li et al., 2018*). In our study, MN was most prevalent among elderly patients, reaching 63.8% in the $\geq$ 60 age group. Across nearly all age groups, the proportion of male MN patients was higher than females, consistent with other reports (*Braden et al., 2000*; *Zhu et al., 2018*). Some studies have identified advanced age and male gender as predictors of progression to end-stage renal disease (ESRD) in MN patients (*Zhu et al., 2018*). Given the rising prevalence of MN, it is crucial to promote early detection through screening for anti-PLA2R antibodies, particularly in elderly populations. Moreover, it is essential for governments to strengthen control over environmental risk factors, such as air pollution, to mitigate the risk of MN.

IgAN was the second most common type of PGN at our center, with a proportion similar to MN, but its proportion is still significantly lower than that in central and northern China. This discrepancy may be attributed to differences in ethnic distribution and age structure across regions (*Xu et al., 2016a*; *Li et al., 2018*). However, IgAN is less common in some countries, potentially due to differences in kidney disease screening practices, as well as genetic and ethnic factors (*Xu et al., 2016b*; *Golay et al., 2013*; *Mubarak et al., 2011*). Studies suggest that the variation in disease prevalence among populations may be linked to differences in allele frequencies, which are shaped by local environmental factors, such as pathogen exposure. This indicates that evolutionary processes could have influenced the global distribution of IgAN (*Magistroni et al., 2015*). Susceptibility gene loci for IgAN have been identified, with protective allele frequencies being lowest among Asians (*Gharavi et al., 2011*). At our center, IgAN was most common among younger patients, accounting for 45.5% of PGN cases in the 14–24 age group and 54.5% in the 25–44 age group. It was also the leading PGN in patients presenting with proteinuria and hematuria, consistent with reports from most other regions of China (*Xu et al., 2016a*).

For SGN, LN was the most prevalent subtype at our center, with a higher proportion among females across nearly all age groups. Therefore, patients with systemic lupus erythematosus should undergo regular monitoring of renal function, urine routine tests, and autoimmune antibody screenings to facilitate early detection and timely intervention of renal dysfunction. Furthermore, personalized treatment plans should be developed for patients already diagnosed with LN.

As the second most common SGN, DN showed a marked increase in proportion during the study period. This trend mirrors findings from other centers in China and internationally, including the US (*Xu et al., 2016a*; *Zheng et al., 2022*; *O'Shaughnessy et al., 2017*). The rise in DN can be largely attributed to the growing prevalence of obesity and diabetes, driven by changes in dietary habits and improved living standards (*Yang et al., 2010*; *Wang et al., 2007*). However, the proportion of DN in our region remains markedly lower than in northwest China. This difference is likely due to the region's preference for a high-salt, high-fat, and high-sugar diet, which is associated with an

increased risk of diabetes and its complications (*Cao et al., 2025*). Additionally, it may also reflect the relatively underdeveloped economy, limited medical resources, and lower health awareness in the northwest (*Zhang et al., 2020*). DN showed the most significant rise during the study period. This underscores the need for stronger diabetes screening and management policies to facilitate early intervention and prevent kidney complications. It is particularly important to promote community-based chronic disease screenings, especially for the elderly. Additionally, enhancing standardized treatment for chronic conditions like diabetes is crucial for better blood glucose control. Screening for the albumin-to-creatinine ratio, estimated glomerular filtration rate (eGFR), and serum creatinine should be more widely implemented, and regular follow-ups should be recommended to track changes in renal function, along with strict adherence to renal biopsy indications and timely treatment to prevent progression to ESRD. Additionally, public health education should be enhanced to raise awareness of the risks about diabetes and its impact on kidney health.

HBVN, by contrast, declined significantly, representing only 0.8% of all renal biopsy cases in 2023, a trend consistent with other studies, likely due to improved public health awareness, education, and vaccination against hepatitis B (*Liang et al., 2009*). The decline in HBVN highlights the success of China's hepatitis B prevention policies. However, continued efforts to improve vaccination rates and promote screening for hepatitis B virus remain critical to further reducing the incidence of HBVN.

HTN is a common renal disease. In this study, only 58 patients had renal biopsy results consistent with HTN, a figure that is significantly lower than its prevalence in the general population. This discrepancy may be attributed to the fact that our center does not routinely perform renal biopsies on patients with chronic HTN unless there was suspicion of concurrent other renal diseases. Additionally, as a tertiary referral institution, our center typically receives patients with more severe and complex conditions, while those with milder cases of hypertension often seek treatment at community hospitals.

Notably, our study period coincided with the global coronavirus disease 2019 (COVID-19) pandemic, which introduced unique challenges in disease patterns that require special consideration. Firstly, during the pandemic, there was a marked decline in health check-ups, and patients with mild clinical symptoms often delayed seeking medical attention. This led to selection bias, which may partially explain the lower proportion of IgA nephropathy (IgAN) observed in our study. Additionally, COVID-19 infection itself can directly or indirectly cause or exacerbate CKD, through mechanisms such as impaired renal perfusion, nephrotoxic drug exposure, increased systemic cytokine production, and endothelial damage (*Brogan & Ross, 2023*; *Legrand et al., 2021*). These factors contribute to a distinct pattern of kidney disease in COVID-19 patients compared to the general population.

This study has several limitations. First, our focus was on the pathological spectrum of kidney diseases, with diagnoses established based on renal biopsy findings and clinical data available at the time of biopsy. However, we did not analyze clinical details during follow-up periods, which limits the ability to establish more comprehensive diagnoses. Second, as a retrospective study, some missing data are inevitable due to the inherent limitations of this research design. Third, our analysis was restricted to patients who underwent renal biopsy, which could introduce selection bias. Patients with less severe

conditions or those who declined the procedure might not be fully represented in this study. Fourth, as a single-center study conducted in a tertiary referral hospital, our findings reflect trends specific to our center, where patients often present with more severe or complex conditions. Patients with milder symptoms are typically managed at community hospitals or primary care settings, making it challenging to generalize our findings to the broader community-level patient population. Lastly, although 95% of our cohort consisted of patients from southeastern China, approximately 5% were from other provinces. This inclusion of non-local patients may slightly limit the generalizability of our findings to trends specific to southeastern China.

## CONCLUSIONS

From 2010 to 2023, the number of renal biopsies at our center increased significantly, with a growing proportion of elderly patients. PGN remained the most common type, with MN gradually overtaking IgAN as the most prevalent subtype. LN was initially the most common SGN, but DN has since become more prevalent. Meanwhile, HBVN cases have steadily declined. These shifts in the kidney disease spectrum provide valuable insights for future public health strategies.

### Funding

This work was supported by Talent Introduction Program of the First Affiliated Hospital of Fujian Medical University (YJRC4003), Young and Middle-aged Scientific Research Major Project of Fujian Provincial Health Commission (No. 2021ZQNZD004). Yanfang Xu was supported by Fujian Research and Training Grants for Young and Middle-aged Leaders in Healthcare (2022QNRCYX-XYF). The funders had no role in study design, data collection and analysis, decision to publish, or preparation of the manuscript.

### Grant Disclosures

The following grant information was disclosed by the authors:
Talent Introduction Program of the First Affiliated Hospital of Fujian Medical University: YJRC4003.
Young and Middle-aged Scientific Research Major Project of Fujian Provincial Health Commission: No. 2021ZQNZD004.
Fujian Research and Training Grants for Young and Middle-aged Leaders in Healthcare: 2022QNRCYX-XYF.

### Competing Interests

The authors declare there are no competing interests.

### Author Contributions

- Zishan Lin conceived and designed the experiments, analyzed the data, prepared figures and/or tables, authored or reviewed drafts of the article, and approved the final draft.

- Shidong Xie conceived and designed the experiments, performed the experiments, analyzed the data, prepared figures and/or tables, authored or reviewed drafts of the article, and approved the final draft.
- Wenfeng Wang performed the experiments, prepared figures and/or tables, authored or reviewed drafts of the article, and approved the final draft.
- Tao Hong performed the experiments, prepared figures and/or tables, authored or reviewed drafts of the article, and approved the final draft.
- Bingjing Jiang performed the experiments, prepared figures and/or tables, authored or reviewed drafts of the article, and approved the final draft.
- Caiming Chen analyzed the data, prepared figures and/or tables, and approved the final draft.
- Xuan Tao analyzed the data, prepared figures and/or tables, and approved the final draft.
- Dewen Jiang analyzed the data, prepared figures and/or tables, and approved the final draft.
- Jianxin Wan analyzed the data, prepared figures and/or tables, and approved the final draft.
- Hong Chen analyzed the data, prepared figures and/or tables, and approved the final draft.
- Yanfang Xu conceived and designed the experiments, authored or reviewed drafts of the article, and approved the final draft.

## Human Ethics

The following information was supplied relating to ethical approvals (i.e., approving body and any reference numbers):

This study was performed in accordance with the Helsinki Declaration and approved by the ethics committee of the First Affiliated Hospital of Fujian Medical University [2015]084-2.

## Data Availability

The raw measurements are available in the Supplementary File.

## Supplemental Information

Supplemental information for this article can be found online at http://dx.doi.org/10.7717/peerj.19302#supplemental-information.

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
