# Peer review of "Changes in the spectrum of kidney diseases: a 14-year single-center retrospective renal biopsy study from southeast China"

_PeerJ, doi:10.7717/peerj.19302_

## Round 0.1 · original submission · Major Revisions

Please respond to the reviewers comments, point by point.

Reviewer 1 ·

Basic reporting

No comments

Experimental design

No comments

Validity of the findings

The weakness of the retrospective data is that it might have missing data. The author didn't describe the checklist used to confirm the diagnosis through the renal biopsy. Also, the p-value provided is not precise for each variable. The p-value is mainly based on comparing the percentage of each kidney disease previously with the current data. Also, the author mentioned exploring their clinical-pathological correlations, but I couldn't see that in the data analysis.

Reviewer 2 ·

Basic reporting

Overall, the manuscript is well-structured and written in clear, professional English.
Some specific suggestions to improve data reporting and visual presentation:
- Figures 3, 4, and 5: Increase the font size for axis labels, percentages, and other annotations to enhance readability.
- Figures 3A and 4A: Consider using the same bar chart style as Figure 2, instead of line charts, for consistency and better comparability.
- Figure 5: Use consistent colors for the same diseases across the chart to improve clarity.
- Figure 5: It would be most interesting to also present the disease proportions for those biopsied due to progressive CKD (in addition to NS and hematuria/proteinuria as you have). This data is crucial, as it aligns with the key focus of your study which is addressing the public health concern of increasing CKD prevalence. Additionally, progressive CKD is your second most common reason for biopsy!

Experimental design

1) Reconsider how you classify kidney disease types. If you wish to keep the "revised 1995 World Health Organization criteria," please cite it explicitly. However, consider updating your definitions to be in line with the more recent 2024 KDIGO GN guidelines. Specific issues with your current definitions:
a) mesangioproliferative GN and MPGN are light microscopy-based histological descriptions rather than discrete disease diagnoses, and now usually considered secondary to some condition rather than a primary GN - unless truly idiopathic. Were these all idiopathic in your cohort?
b) HBVN - how did you define this? HBVN can have variable histological appearances like membranous or MPGN. (Again, see KDIGO 2024 GN guidelines) Did you do HBV staining on the biopsy tissue? If so, provide methodological details about how your lab conducts HBV tissue stains.
c) How were IgAN and HSPN (which is now typically referred to as IgA vasculitis) differentiated? They are identical on renal pathology, and differentiating them requires chart review for other disease manifestations - so did the two nephrologists who reviewed the pathology also review clinical charts?

2) Related to 1c above, clarify your diagnostic process overall.
- Were the diagnoses were based solely on histopathology? If this is the case, then again see above issue with IgAN vs IgAV)?
- Were the diagnoses determined after on independent chart review by the 2 nephrologists? If this is the case, what was the process for settling diagnostic disagreements?
- Or were the diagnoses simply those assigned by the treating clinician during the patient encounter?

3) As mentioned in the above comment on Figure 5 - progressive CKD accounted for 22.4% of your biopsies, making it the second most common reason for biopsy in your cohort. It would be important to provide more details on the underlying causes identified in this group.

4) Hypertensive nephropathy, known to be extremely common, is noticeably absent from the listed diseases. Why is this - does this reflect a local practice pattern where patients with chronic HTN are typically not biopsied? Please address this in your conclusions.

5) Is electron microscopy typically done for your center's kidney biopsies? EM may not be widely available in many regions, but its contribution to GN diagnoses is important when it's available - so please specify whether (and if any, how many) samples were analyzed with EM.

6) As background for an international readership that may not be familiar with your geographic area, it may be helpful to tell us more about your medical center - eg. a sentence on the size of population you serve, whether you are a tertiary referral center, etc.

7) Corrections needed in the second sentence of the introduction ("In China, the INCIDENCE of CKD has grown from 6.7% to 10.6% over the past three decades, and MORTALITY RATES have increased from 8.3 to 13.8 per 100,000 people.")
- 6.7%, 10.6% are PREVALENCE and not incidence
- "mortality rate" is per unit time, not just a frequency

Validity of the findings

The validity of the findings would be enhanced with clearer and more standardized classification of kidney disease types, as outlined above.

Nevertheless, the conclusions remain meaningful and carry significant public health implications, particularly emphasizing the critical need for interventions targeting diabetes.

Additional comments

Overall, the authors make a valuable contribution to understanding the changing epidemiology of CKD in their geographic region. The findings have important implications for public health interventions, particularly in targeting diabetes. With some refinements as outlined above, the study’s findings could be further strengthened.

Reviewer 3 ·

Basic reporting

While this study provides valuable insights, the claim that it represents southeastern China is overstated for a single-center study. If the authors aim to generalize their findings to this region, it is crucial to provide more specific inclusion and exclusion criteria. For instance, how were patients from outside the local area, such as those undergoing biopsies while visiting family or traveling, accounted for in the analysis? Clarification on this matter would enhance the reliability of the findings.

The discussion primarily reiterates the results without providing sufficient interpretation or addressing the research questions posed in the introduction, particularly regarding regional differences within China. The introduction suggests that geographic variation exists in kidney disease prevalence and patterns, yet the discussion lacks an in-depth examination of how the study's findings align with or diverge from this premise. Including a comparative analysis with data from other regions in China would enhance the discussion's relevance.

The statement, "These differences may be attributed to variations in genetic predisposition, environmental exposure, and kidney disease screening policies," is overly general and avoids addressing the key implications of the study. The authors should elaborate on how changes in kidney disease screening policies might influence disease detection and outcomes. For instance, what specific public health strategies could be implemented to address the increasing trends of MN and diabetic nephropathy (DN) while mitigating the observed decrease in HBVN? A deeper exploration of these questions is essential to substantiate the conclusion.

The conclusion highlights the need for "ongoing public health efforts tailored to the changing spectrum of kidney diseases." However, the study does not provide specific recommendations on how these efforts should be tailored. For instance, what screening or management strategies should be prioritized based on the observed trends? Providing concrete suggestions would enhance the study's contribution to public health policy and practice.

Experimental design

The study does not clearly distinguish between primary and secondary membranous nephropathy. Given the mention of hepatitis B-associated nephritis (HBVN), it is likely that secondary MN cases are included in this category. A detailed classification of MN into primary and secondary forms would add depth to the analysis and avoid potential misinterpretations of the data.

Validity of the findings

The study period includes the COVID-19 pandemic, which likely affected biopsy practices. For example, mild urinary abnormalities may have been under-investigated during this time. This could partially explain the relatively low proportion of IgA nephropathy cases. The authors should discuss how the pandemic may have influenced the study's findings and consider adjusting the analysis to account for this potential bias.

Despite the limitations noted above, the dataset is valuable and provides important insights into the changing spectrum of biopsy-confirmed kidney diseases over a 14-year period. A more detailed exploration of these trends and their implications could significantly strengthen the manuscript.

---

## Round 0.2 · accepted · Accept

All the comments were well addressed.

Reviewer 3 ·

Basic reporting

The author's adequately addressed my comments in the text.

Experimental design

The author's adequately addressed my comments in the text.

Validity of the findings

The author's adequately addressed my comments in the text.